# Massively parallel jumping assay decodes *Alu* retrotransposition activity

Navneet Matharu[1,2,10] ✉, Jingjing Zhao[1,2,10], Ajuni Sohota[1,2], Linbei Deng[1,2], Yan Hung[1,2], Zizheng Li[1], Kelly An[1,2], Jasmine Sims[1,2], Sawitree Rattanasopha[1,2], Thomas J. Meyer[3,4], Lucia Carbone[3,5,6,7], Martin Kircher[8,9] ✉ & Nadav Ahituv[1,2] ✉

The human genome contains millions of copies of retrotransposons that are silenced but many of these copies have potential to become active if mutated, having phenotypic consequences, including disease. However, it is not thoroughly understood how nucleotide changes in retrotransposons affect their jumping activity. Here, we develop a massively parallel jumping assay (MPJA) that tests the jumping potential of thousands of transposons *en masse*. We generate a nucleotide variant library of four *Alu* retrotransposons containing 165,087 different haplotypes and test them for their jumping ability using MPJA. We found 66,821 unique jumping haplotypes, allowing us to pinpoint domains and variants vital for transposition. Mapping these variants to the *Alu*-RNA secondary structure revealed stem-loop features that contribute to jumping potential. Combined, our work provides a high-throughput assay that assesses the ability of retrotransposons to jump and identifies nucleotide changes that have the potential to reactivate them in the human genome.

An estimated 42% of the human genome consists of retrotransposable elements that "copy and paste" in genomes via RNA-mediated transposition[1,2]. One of the most abundant classes of retrotransposons in mammalian genomes is "*Alu*" elements that are derived from 7SL RNA. *Alu* elements belong to the class of "Short Interspersed Nuclear Elements" that have an Alu-1 restriction endonuclease site derived from the *Arthrobacter Luteus* bacteria. *Alu* elements are non-autonomous retrotransposons, as they borrow transposition machinery from other retrotransposons (i.e., L1-ORF2p) to jump[3]. There are roughly 1.1 million *Alu* copies in the human genome[4], which can be divided into three major subfamilies: (1) *Alu*J, the most ancient (~65 million years old, myo) and whose sequence is substantially degraded

and thus believed to be inactive[3]; (2) *Alu*S, estimated to be ~30 myo and has elements that are less diverged from the consensus sequence and believed to include some active elements in the human genome; (3) *Alu*Y, is present in old world monkeys at least ~25 myo ago. Some of the recent (~10 myo) *Alu*Y elements are thought to be active in the human genome. The human genome is predicted to have 852 intact functional *Alu* elements with thousands of copies that could be competent to jump[5], and it is estimated that a new *Alu* insertion happens in the human genome every 40 live births[6]. The jumping activity of *Alu* elements positively correlates with the length of their polyA tail[7–9]. Additionally, the presence of an intact 280 base pair (bp) core *Alu* sequence is thought to be essential for retrotransposition[9].

[1]Department of Bioengineering and Therapeutic Sciences, University of California San Francisco, San Francisco, CA, USA. [2]Institute for Human Genetics, University of California San Francisco, San Francisco, CA, USA. [3]Division of Genetics, Oregon National Primate Research Center, Beaverton, OR, USA. [4]CCR Collaborative Bioinformatics Resource (CCBR), Frederick National Laboratory for Cancer Research, Leidos Biomedical Research Inc., MD Frederick, USA. [5]Department of Medicine, Knight Cardiovascular Institute, Oregon Health and Science University, Portland, OR, USA. [6]Department of Molecular and Medical Genetics, Oregon Health and Science University, Portland, OR, USA. [7]Department of Medical Informatics and Clinical Epidemiology, Oregon Health and Science University, Portland, OR, USA. [8]Berlin Institute of Health of Health at Charité—Universitätsmedizin Berlin, Berlin, Germany. [9]Institute of Human Genetics, University Medical Center Schleswig-Holstein, University of Lübeck, Lübeck, Germany. [10]These authors contributed equally: Navneet Matharu, Jingjing Zhao. ✉e-mail: nkmatharu@gmail.com; martin.kircher@uni-luebeck.de; nadav.ahituv@ucsf.edu

Comprehensive sequence homology analysis of the 280 bp core region of 89 *Alu* elements from the human genome revealed that at least 124 nucleotide positions are conserved in all *Alu* elements that were tested to be active in retrotransposition assays[5]. However, no exhaustive functional study has been performed to date to correlate *Alu*-jumping activity with sequence alterations. A high-resolution sequence-based functional analysis of the 280 bp *Alu* core sequence is challenging, mainly due to the lack of high-throughput retrotransposition assays. An assay for measuring the jumping activity of individual *Alu* elements has been established, utilizing splicing during the jumping event to infer antibiotic resistance, but can only test one element at a time[5]. In addition, while there have been studies that retrieve libraries of polymorphic LINE1 retrotransposons from the genome, directed evolution retrotransposition assays for *Alu* elements are lacking[10,11].

Here, we developed a massively parallel jumping assay (MPJA) that can test in parallel the ability of thousands of *Alu* elements to retrotranspose in human cells (Fig. 1). We used it to test the reactivation potential of three different inactive *Alu* elements and one active element. Using error-prone PCR, we generated 165,087 different *Alu* haplotypes and tested their jumping ability in HeLa cells. We found 66,821 haplotypes that enabled jumping and used these results to characterize functional *Alu* domains and annotate variants that are vital for retrotransposition. We observed that even a single-nucleotide change is sufficient to change the activity of an *Alu* element. Overlapping these jumping-associated variants with *Alu*-RNA secondary structure identified stem-loop features that contribute to retrotransposition potential. In summary, our results provide a technology that can test the jumping ability of thousands of sequences and identify key domains, RNA secondary structures, and variants that are vital for retrotransposition.

## Results

### Retrotransposition assay optimization

We developed an MPJA, based on a previous assay for individual *Alu* transposition[12]. In this assay, the retrotransposition cassette includes an *Alu* transcript that is driven by a 7SL Pol*III* enhancer (Fig. 1b). The 3′ of the *Alu* transcript has a Pol*II* driven SV40-Neomycin (*Neo*) cassette cargo in antisense orientation before a Pol*III* termination signal, i.e., polyT tract. Upon Pol*III Alu* transcription, the resulting RNA gets spliced, due to the presence of an autocatalytic intron that is independent of Pol*II* splicing and engages with the L1-ORF2 transposon machinery, which is needed for its genomic integration. The integrated *Alu* copy loses 7SL and the Pol*III* polyA retains the Pol*II* SV40-spliced *Neo* cassette, providing neomycin resistance for colonies that undergo a retrotransposition event, which can then be used for selection.

We first set out to standardize the retrotransposition assay, using *Alu*Ya5, a highly active *Alu* element[12], as a positive control. We used the HeLa-HA subline that is known to be permissive for this *Alu* mobilization assay[13,14]. A vector containing *Alu*Ya5 was co-transfected with L1-ORF2 into HeLa-HA cells (Fig. 1b). These cells were replated once before starting G-418 selection 4–5 days post-transfection. After 3 weeks of G-418 stable selection, sizable colonies were observed (Fig. 1b, Supplementary Fig. 1 showing colony assay). Of note, we also observed a small number of colonies in the negative control (non-L1-ORF2), likely due to random integration of the *Alu*Ya5 plasmid. These random unspliced product integrations could be easily detected by PCR primers unique to the vector and thus can be readily excluded from further processing. All colonies, both from *Alu*Ya5 and the negative control, were further processed for genomic DNA extraction to examine *Alu* transposition. Upon splicing, the amplicon should be 500 bp shorter than the unspliced 1.5 kb band, allowing to distinguish random integration events versus transposition events (Fig. 1c). We found *Alu*Ya5 colonies to have a 500 bp shorter PCR product compared to zero colonies from the negative control (Supplementary

Fig. 2a, Source Data Fig. 1a). This jumping assay is therefore robust in detecting *Alu* sequences that lead to retrotransposition.

### Selection of *Alu* elements for MPJA

Having optimized the retrotransposition assay, we next set out to choose a library of *Alu* elements to be tested by MPJA. A previous study that assessed the jumping activity of various *Alu* families from the human genome found the *Alu*J family to be inactive and have a highly divergent 280 bp core region, the *Alu*S family to be moderately active, and the *Alu*Y family to be highly active, both having an intact 280 bp core[5]. For our MPJA, we wanted to test what nucleotide changes are required to reactivate inactive *Alu* elements that have an intact 280 bp core. We selected members of the *Alu*S family, since the AluS family of retrotransposons largely consists of inactive elements with an intact 280 bp core and less diverged from the consensus *Alu*Sx sequence that is active in the retrotransposition assay[5]. A previous study that assessed 26 full-length *Alu*S elements found 21 to be inactive, having low to zero activity[5]. Building on these results, we applied stringent filters of inactivity to the same subfamily, to select three *Alu*S inactive candidates that have an intact 280 bp core region and at least 95% sequence similarity with the consensus active *Alu*Sx sequence: *Alu*S-6b, *Alu*S-14b and *Alu*S-h1.1 (Fig. 2a). To further validate that they are indeed inactive or have very low-jumping potential, we tested them individually in our assay confirming their activity as previously shown in a retrotransposition assay[5] (Supplementary Fig. 3).

### *Alu*-MPJA saturation mutagenesis

We synthesized all four *Alu* sequences and cloned them into the *Alu*-jumping vector, followed by Sanger sequence validation. We then used an error-prone PCR system that is estimated to generate a mutation every 1–16 bp per kilobase (kb) of DNA (see "Methods") to generate roughly 1–6 mutations per 300 bp *Alu* element[15,16]. The mutagenized PCR product of each *Alu* element was then re-cloned into the *Alu*-jumping assay vector to generate four mutagenized libraries: *Alu*Sx-mut, *Alu*6b-mut, *Alu*14b-mut, and *Alu*h1.1-mut (Supplementary Data 1). To assess their complexity, libraries were subjected to two rounds of massively parallel sequencing, finding both replicates to show a high degree of overlap and correlation in count representation of identical variants (Pearson and Spearman correlation coefficiencies, *Alu*Sx p-rho 1.00 s-rho 0.8; *Alu*6b p-rho 1.00 s-rho 0.72; *Alu*14b p-rho 1.00 s-rho 0.77; *Alu*h1.1 p-rho 1.00 s-rho 0.77; Supplementary Fig. 4a). We found all four libraries to be highly saturated for single-nucleotide variants (SNVs) with frequencies of transitions, transversion and indels at each position (Supplementary Data 2, Supplementary Fig. 5). On average, we detected 10 variants in each individual *Alu* sequence, due to a small subset of variants that were fixed in the early error-prone PCR amplification, leading to a "founder effect" (Supplementary Fig. 5). We also observed that nucleotide positions 1–5 in *Alu*Sx, *Alu*6b and *Alu*14b libraries were never mutated. This could be attributed to our random mutagenesis strategy that includes a 5′ primer overlapping up to 20 terminal nucleotides (Supplementary Fig. 10). For *Alu*h1.1, base positions 1–5, 9, 10, 283, and 284 were never mutated. We also observed that there was no nucleotide position that was 100% mutated, i.e., created all possible mutations for that position, including SNV and INDELS.

We next performed the retrotransposition assay with all four *Alu*-Mut libraries. Libraries were transfected into HeLa cells, which were then treated with G-418 three days post-transfection. Following 3–4 weeks of G-418 selection, colonies were amplified and replated to achieve confluency and later subjected to genomic DNA extraction. Using specific PCR primers (Supplementary Data 3), we amplified all the *Alu* sequences from the genomic DNA that were retrotransposed. First, we specifically amplified the 1 kb spliced Neo-*Alu* amplicon to differentiate from any plasmid integrations that could happen without retrotransposition, allowing to resurrect only transposed *Alu* elements from the genome

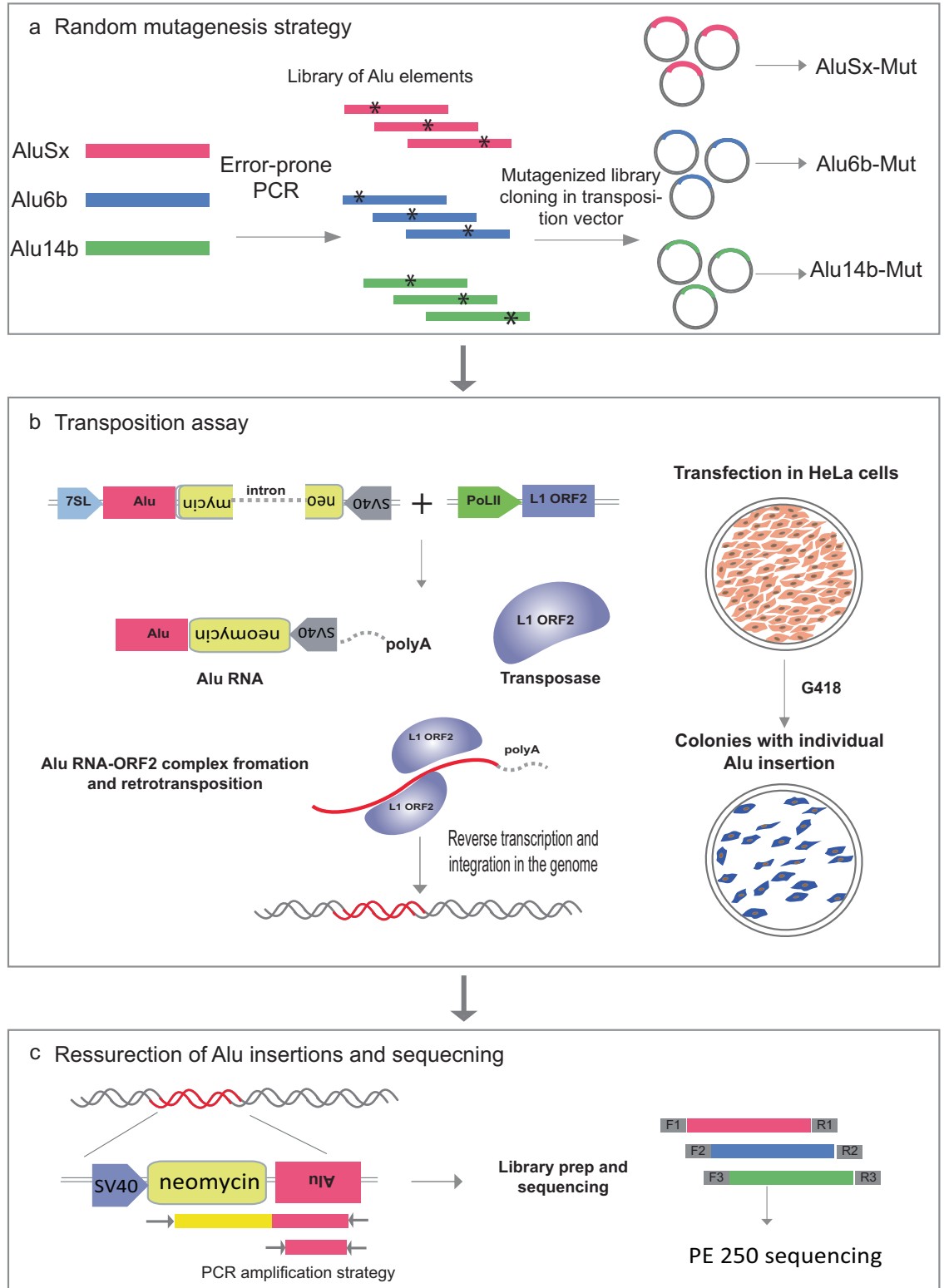

**Fig. 1 | Massively parallel jumping assay. a** Schematic showing the random mutagenesis strategy to generate *Alu* variant libraries using error-prone PCR. **b** Retrotransposition assay and *Alu* integration into the genome with the help of L1 transposase machinery. The transposition vector contains RNA Pol*III* 7SL driving an *Alu*-Neo cassette that is spliced and complexed with the ORF2 transposase machinery from the helper plasmid. The *Alu*-Neo cassette gets reverse transcribed and integrates randomly into the genome, allowing the neomycin resistance gene to be expressed through the RNA Pol*II* SV40 promoter, thus conferring G-418 sulfate-resistant colonies. **c** Retrotransposed *Alu* resurrection and retrieval from the genome and sequence library generation. *Alu*-Neo integrations were selected using neomycin-specific and *Alu*-specific primers that generate a 1 kb PCR product if neomycin is spliced due to retrotransposition. *Alu*-specific primers are then used to amplify the integrated *Alu*s, which are then processed for sequencing.

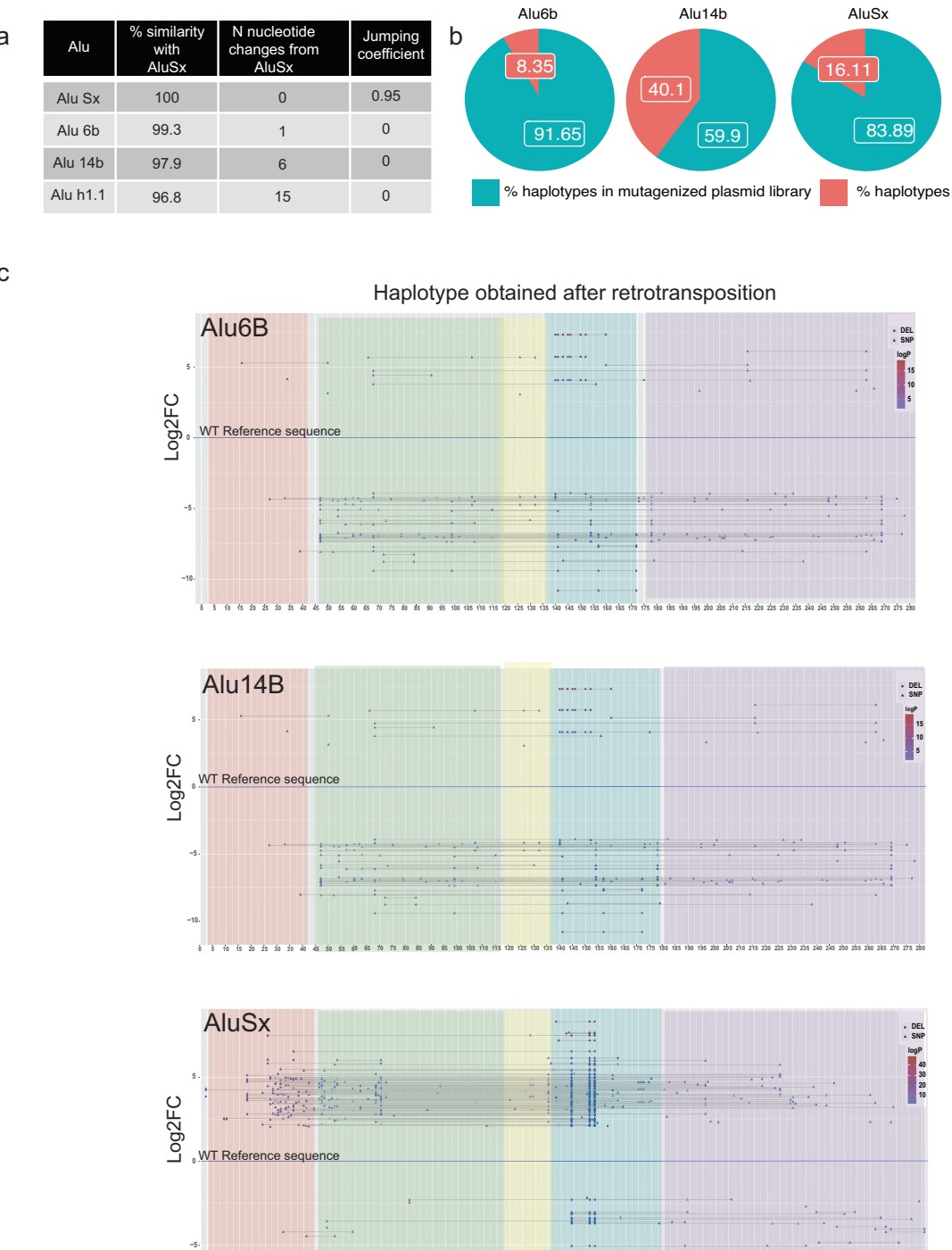

**Fig. 2 | *Alu* selection and variants calling in the mutagenized library. a** Sequence similarity (percent sequence identity), the number of sequence differences, and the relative jumping potential to *Alu*Ya5 (positive control element considered to have activity score of 1) for all four tested *Alu* sequences. **b** Pie charts showing the percent of haplotypes detected in the *Alu*-Mut jumping libraries from total haplotypes detected in *Alu*-Mut plasmid libraries. **c** Variant calling at each position across the 280 bp full-length *Alu* sequence in the *Alu*-mutagenized jumping/plasmid library and haplotype calling depicting only significant high-jumping and low-jumping haplotypes beyond a cutoff of ±2.5 log2 fold-change (Log2FC). Significant jumping was defined via the DESeq2 package[17] using a Wald-test (two-sided) *p* value threshold of 10[−5]. The lower panel shows a schematic depicting the major structural features in *Alu*-RNA.

(Supplementary Fig. 2b, Source Data Fig. 1b). We also checked for primer set specificity of the Neo-Alu amplicon primers for any non-specific amplifications from the genomic DNA of untransfected HeLa cells (Supplementary Fig. 2c). Next, we used primer targeting the vector sequence on 3′ of the cloned *Alu* sequence and 5′ *Alu*-specific primers to amplify 300 bp fragments containing 280 bp *Alu* core sequence (Methods). We then generated a sequencing library from the resulting ~280 bp band and indexed and pooled libraries in equimolar ratios for sequencing.

### *Alu*-MPJA saturation mutagenesis analysis

We next analyzed the *Alu*-Mut library (the plasmid variant library) and *Alu*-jumping library (the library following retrotransposition) for each *Alu* element. We first obtained the total number of read counts for single-nucleotide variants in each plasmid (*Alu*-Mut) and *Alu*-jumping library. We observed that most of the variants fall within the mismatch window of up to <10 nucleotide changes for all four *Alu* elements (Supplementary Fig. 6) in both *Alu*-Mut and *Alu*-jumping libraries. We then calculated log2 fold-change (log2FC) based on the frequency of reads from the *Alu*-jumping vs *Alu*-Mut library. The significant variants with a positive fold-change and negative fold-change were determined using the DESeq2 R package[17] (Supplementary Fig. 5). We normalized fold-changes with the measured activity of the reference *Alu* sequence that we started our mutagenesis on to provide a "jumping score" for the sequences (Fig. 2a). We observed that highly enriched SNVs in the *Alu*-Mut plasmid library, either due to PCR bias or founder effect, were not necessarily enriched in the *Alu*-Mut jumping library, confirming that our measurements are focused on jumping activity (Fig. 2c, Supplementary Fig. 10). As mentioned earlier, the PCR mutagenesis strategy is non-random and can lead to founder mutation effects. However, we found many different SNVs in the SRP9-14 binding region of *Alu*Sx-Mut sequences in the plasmid library, suggesting that we created different haplotypes at "founder positions" although with different frequencies. After jumping, these haplotypes underwent retrotransposition, and depending upon their ability to jump, these haplotypes were enriched or depleted in the Alu-Mut jumping libraries.

Next, we compared results from the two *Alu*-Mut jumping technical replicates (Supplementary Fig. 4b), finding them to show good correlations (Pearson and Spearman correlation coefficients, *Alu*Sx p-rho 1.00 s-rho 0.59; *Alu*6b p-rho 0.95 s-rho 0.60; *Alu*14b p-rho 1.00 s-rho 0.54) except for *Alu*h1.1, which had a poor correlation between replicates (*Alu*h1.1 p-rho 0.44 s-rho 0.16) (Supplementary Fig. 4b). We thus decided to remove *Alu*h1.1 from all subsequent analysis. We also screened our *Alu*-Mut plasmid libraries for the possible phenomenon of index hopping reported earlier on Illumina sequencing platforms[18]. Despite using dual-indexing, which was reported previously to be able to address the inaccuracies of multiplexing, we did observe a low proportion of potential index hopping (for *Alu*Sx 0.8%, *Alu*6b 2.3%, *Alu*14b 3%, total 2.2% in *Alu*-Mut plasmid libraries)[19]. We note that this might also be caused by spurious sharing of sequences or could be the result of cross-contamination before library prep, and we removed all these reads in our subsequent analysis.

We annotated *Alu* haplotypes, requiring a limit of up to 10 nucleotide changes per element, as we wanted to focus on the minimum number of sequence variants required to change the activity of the *Alu* element. In addition, we wanted to prioritize those haplotypes in line with the low rate of sequence variants created by the mutagenesis strategy that we adopted, which typically allows 1–16 base changes per kb according to the Genemorph mutagenesis kit (see "Methods"). We called a haplotype only if it had read counts higher than two in the plasmid library and greater than ten in the jumping library. In total, we annotated 20,733, 48,641, and 35,849 haplotypes in *Alu*Sx, *Alu*6b, and *Alu*14b, respectively, in the plasmid library. Of these 9493, 10,463, and 32,176 haplotypes were found in

the jumping libraries of *Alu*Sx, *Alu*6b, and *Alu*14b, respectively (Fig. 2; Supplementary Data 4–7). These numbers include both high and low jumpers without any further cutoffs. We found that the aforementioned highly enriched SNVs in the plasmid libraries, which likely appeared due to PCR bias and founder effects during library generation, were not significantly enriched in *Alu*-jumping libraries (with ±2 log2FC, $p$ value < $10^{-5}$), suggesting that they have a minimal effect on determining jumping activity (Fig. 2c, Supplementary Fig. 10). Interestingly, we also observed in the *Alu*-jumping libraries some fragments (>20–25 Mismatches) that were shorter than 200 bp that were missing the right arm monomer (Supplementary Fig. 7), fitting with previous reports that showed that *Alu* elements missing this arm can still retain jumping activity[20]. We checked where the reads start (Supplementary Fig. 8) and all these <200 bp fragments were clustered at around the same position in the *Alu* after the left monomer (Supplementary Fig. 7). We confirmed via PCR amplification using *Alu* recovery primers (Supplementary Data 3) that these short reads are not an artifact due to non-specific amplification from the untransfected HeLa cell genome (Supplementary Fig. 2c). We removed these haplotypes from our subsequent analyses.

### Identification of haplotypes that alter *Alu*-jumping potential

We next set out to identify *Alu* haplotypes that lead to a significant increase in retrotransposition compared to their reference wild-type *Alu* sequence, by measuring differential enrichment in *Alu*-Mut jumping vs *Alu*-Mut plasmid libraries (Figs. 2c and 3a–c, Supplementary Data 4–6). For each library, the normalized counts of all haplotypes were calculated using DESeq2[17]. We estimated log2FC by comparing Jumping haplotypes to the Plasmid library (Methods) and normalizing to the reference *Alu* sequence (Fig. 3a–c, Supplementary Data 4–6). We applied a cutoff of Log10P $p$ value threshold of $10^{-5}$ for significant haplotypes and refer to >2 log2FC as high jumpers and <2 log2FC as low jumpers, respectively (Fig. 3a–c, Supplementary Data 12). Haplotypes that were observed in the plasmid library but not in the jumping library we referred to as non-jumpers. We observed that the majority of the unclassified haplotypes that did not fall under any of the above classes (Fig. 3a, b, Supplementary Data 12) for *Alu*6b and *Alu*14b have close to or lower log2FC than their reference sequence, signifying their inherent low-jumping potential. In contrast, for *Alu*Sx, which is inherently active, there were more non-classified haplotypes that showed higher log2FC (Fig. 3c, Supplementary Data 4). Due to the difference in their inherent jumping activity, we compared haplotypes within each *Alu* library and not across them. We also generated volcano plots for the haplotypes with non-normalized results to the *Alu* reference WT sequence (Supplementary Fig. 11). We next analyzed the number of nucleotide changes required to obtain jumping activity. We found that a minimum of two changes are needed for *Alu*6b and five for *Alu*14b. For *Alu*Sx, as it is already an active jumper, we analyzed how many changes are needed to increase its jumping potential, finding that one nucleotide change was sufficient to substantially increase the >2 log2FC (Fig. 3d, Supplementary Data 4–6).

Our assay only detects variants that are present in *Alu* sequences that jumped, but does not detect haplotypes that are either non-jumpers or could be absent due to other reasons that are not related to retrotransposition ability. We thus applied a significantly higher threshold of read counts for potential haplotypes that reduce jumping activity, requiring over 50 reads in the plasmid and zero reads in the jumping library in any of the replicates. This allowed us to separate out potential non-jumpers (Fig. 3a–c) from background that could be due to various experimental effects. Combining results of the low and non-jumpers, we found that one nucleotide changes for *Alu*Sx, two nucleotide changes for *Alu*6B, and seven nucleotide changes for *Alu*14B were required to decrease or abolish jumping potential (Fig. 3d).

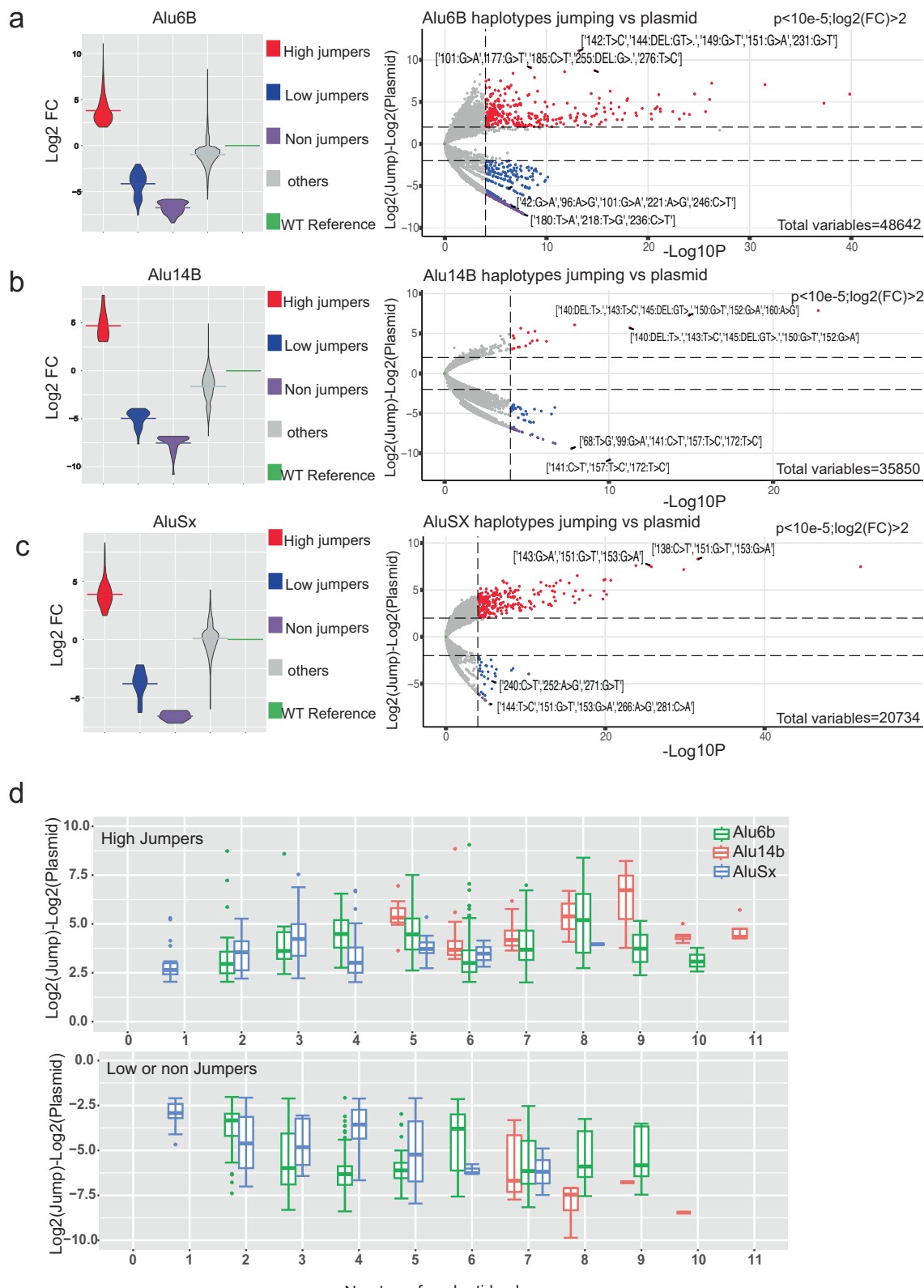

## Mutations that affect jumping are associated with SRP binding domains

We next analyzed the dataset along the length of the *Alu* element to reveal positions or domains where mutations have the potential to contribute towards positive or negative jumping activity. We used a 5 bp sliding-window analysis to score the activity from different nucleotide variants over the sliding window across the full length of the *Alu* element (Fig. 4). We fit multiple linear regression models and plotted the coefficients reflecting the combined effect of variants over the respective 5 bp tiles/windows. Our results indicate that the log2FC haplotype activity presented above correlates well with the 5 bp window analysis, which allows us to compensate for the differences in

**Fig. 3 | Haplotypes of dominant jumping effects. a–c** Violin (left panel) and Volcano (right panel) plots showing fold-change differences in high jumpers, low jumpers, and possible non-jumping *Alu* haplotypes with cutoff of ±2 log2 fold-change (Log2FC) for *Alu*14B-Mut (**a**), *Alu*6B-Mut (**b**), *Alu*Sx-Mut (**c**), respectively. Significant effects were defined using the DESeq2 package[17] with a Wald-test (two-sided) and *p* value threshold of $10^{-5}$. Classes were defined as significant high jumpers in red (Log2FC > 2), significant low jumpers in blue (Log2FC < −2, jumping counts > 0), and non-jumpers in purple (Log2FC < −2, jumping counts = 0 and plasmid count > 50). Non-significant haplotypes are shown in gray. Plots are normalized to a zero Log2FC of the reference or wild-type sequence (in green) of each element. Horizontal lines in the Violin plot are the median. **d** The number of nucleotide changes observed in the library with respect to the reference or wild-type sequence for high jumper (top) and low/non-jumper haplotypes (bottom). Box and whisker plots include the median line, the box denotes the interquartile range (IQR), whiskers denote the rest of the data distribution, and outliers are denoted by points greater than ±1.5 × IQR.

variant density and representation of certain variants from the saturation mutagenesis process.

*Alu* retrotransposons are transcribed by the RNA Pol*III* machinery and fold into a specific RNA structure[21]. This secondary structure of *Alu*-RNA is crucial to engage the transposition machinery[20,22]. *Alu*-RNA has distinct regions in its left half (1–120 bp), termed the left arm or monomer, followed by a middle A-stretch and a right arm or monomer (150–300 bp) (Fig. 2c). The left and right arms of *Alu*-RNA fold independently in a series of stems and loops. One of the secondary structures that is formed at the 5′ of each left and right arm is the bipartite stem-loop structure that binds to the signal recognition particle (SRP9/14) complex. The SRP binding site on the left arm has annotated BoxA and BoxB regions that are known to be crucial for retrotransposition[9,23,24]. SRP complex binding is crucial for the process of transposition and nucleotide changes that alter the binding affinity of the SRP complex can affect the transposition efficiency[22]. The 5 bp window analysis clearly indicated an enrichment of activity alteration in the SRP binding domains that correspond to *Alu*-RNA secondary structure.

We used RNA fold to predict the secondary structure of the *Alu*-RNA[25]. We analyzed whether nucleotide changes in regions that were predicted to positively affect the haplotype frequency are enriched in the SRP binding stem-loop structure (Fig. 4, Supplementary Data 4–6 and 11). We performed a hypergeometric test of variants in high jumpers and low/non-jumpers in the SRP binding regions for both left and right SRP binding structures. Interestingly, we observed that variants in the SRP binding stem-loop structures were highly enriched for high jumpers compared to low/non-jumpers in all *Alu* classes tested (Hypergeometric test *Alu*6b *P* value = 0.00, Alu14b *P* value = 1.65e-05, *Alu*Sx *P* value = 0.00) (Fig. 4). We performed sequence analysis on both left and right SRP binding structures to pinpoint changes that are contributing towards Alu-jumping potential (Fig. 4, Supplementary Fig. 9, Supplementary Data 11). We found that individual nucleotide alterations in these regions were sufficient to modify *Alu*-jumping activity. Furthermore, we observed clear differences in the frequency of certain nucleotides at specific positions in high-jumping vs low-jumping haplotypes. We compared the nucleotide changes that occur in high versus low jumpers. For example, position 153 shows different mutations in both low and high jumpers that were not the founder SNV mutation of the plasmid library. There are many haplotypes observed (with different SNVs) at position 153, suggesting that there was no bias from the founder mutation in the plasmid library during jumping. We report the frequency of each nucleotide in each SRP domain in the left and right arm separately for low and high-jumping Alu haplotypes (Supplementary Data 11).

We validated these results using a standard retrotransposition colony assay by testing two high and two low-jumping haplotypes for *Alu*Sx, having mutations in their SRP domains, finding similar results to the MPJA (Supplementary Fig. 12). In summary, our results show that nucleotide changes overlapping predicted *Alu* SRP binding regions have a major effect on transposition.

### Comparison to *Alu*S sequences in the human reference genome

*Alu*S family of elements in the human genome is inactive due to sequence changes. We next set out to quantify how many mutations

could lead to the activation of current *Alu*S elements in the human reference genome if mutated. The human genome has around half a million (551,383) copies of *Alu*S elements, and around 852 of those have an intact 280 bp core[5]. We extracted *Alu*S elements previously identified from the genome[5] and added a custom identifier for each (Supplementary Data 8 and 9). We first aligned our *Alu*S jumping haplotypes to the subset of *Alu*S sequences in the human genome that have an intact 280 bp core region, finding that none of these mutagenized haplotypes have an exact match with these elements. This was expected, as previously reported by testing 27 *Alu*S elements individually, that no full-length *Alu*S is active in a retrotransposition assay when it is 10% divergent (28 nucleotide changes) from the consensus active *Alu*Sx sequence[5]. We first plotted the total number of haplotype counts generated in the *Alu*-Mut jumping libraries by the number of mismatches to existing *Alu*S sequences in the human genome with an intact 280 bp core (Fig. 5a, Supplementary Data 8 and 9). From our MPJA study, we found that the closest jumping mutagenized haplotypes for *Alu*6B or *Alu*Sx are seven mismatches away from endogenous *Alu*S and ten mismatches for *Alu*14B (Fig. 5a, b, Supplementary Data 8 and 9). Analyzing only the high jumper haplotypes, we found eight mismatch differences for *Alu*6B or *Alu*Sx and twelve mismatches for *Alu*14B (Fig. 5b, Supplementary Data 8 and 9). We also looked at how many *Alu*S elements in the human genome could potentially become jumpers upon n number of mismatches. We plotted the number of genomic *Alu*S that have mismatches with only high jumper haplotypes from our mutagenesis library. We found up to fifteen high-jumping haplotypes in our library similar to the eight distinct *Alu* elements from the genome that are 8–16 mismatches away from being high jumpers (Fig. 5c, Supplementary Data 8 and 9). In our assay, we found that some of the currently present *Alu* elements in the human genome could become active if they incorporate 8–16 mutations.

## Discussion

Retrotransposition of genomic elements is a major force driving genomic diversity[1]. Around 11% of the human genome consists of *Alu* retrotransposons, with 1.4 million copies existing in various genomic locations. With a germline mutation rate of $1.2 \times 10^{-8}$ per base pair per generation[26–28], the sequences of many *Alu*s can change over time, which could potentially lead to their reactivation. Most *Alu* elements are inactive due to sequence disintegration, but a few have maintained sequence features necessary for jumping, i.e., an intact 280 bp core sequence with BoxA and BoxB in the left monomer as well as a PolyA tail[9]. It is unknown how nucleotide changes present throughout the *Alu* elements may affect their jumping potential. The assays developed so far measure the activity of an individual transposon, precluding our ability to identify nucleotide changes at a higher resolution level that could affect their jumping potential. To address this, we developed an assay that can test in a high-throughput manner the jumping capability of thousands of sequences. We used four different *Alu*S elements, constructing libraries containing 167,534 unique haplotypes and tested their jumping ability. We tested different unique haplotypes for each Alu element and compared them with their reference wild-type version to record their jumping potential. This assay could be easily modified for other *Alu* families,

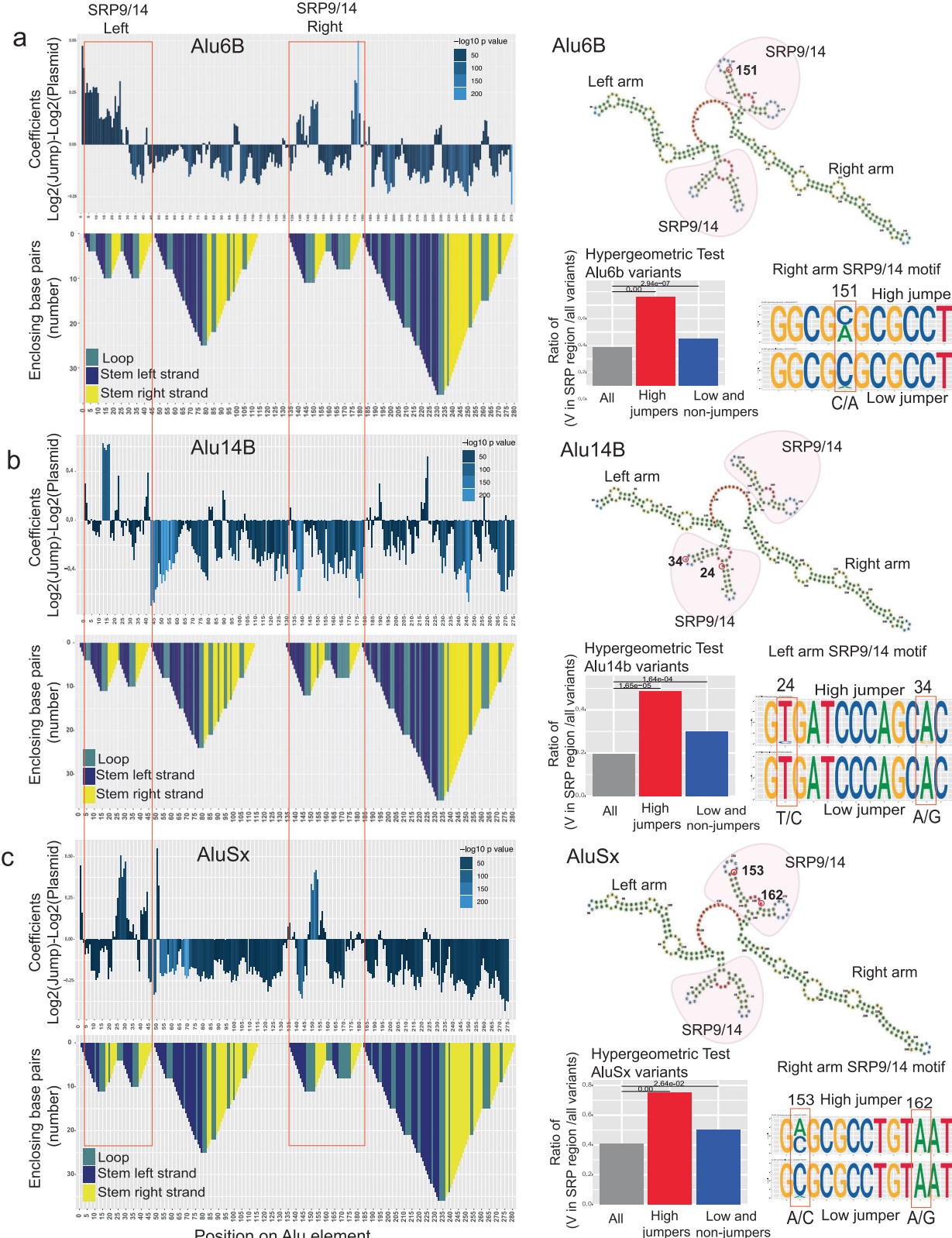

**Fig. 4 | Mutations that affect jumping are associated with SRP binding domains. a–c** In the top panel, 5bp-sliding-window analyses of *Alu*6B (**a**), *Alu*14b (**b**), and *Alu*Sx (**c**) are shown along with the *Alu*-RNA folding structure in the panels below. The folding structure is reverse mountain coded with different colors for each secondary structure: stem-loop-unhybridized in teal, hybridized left strand of the stem in dark blue, and hybridized right arm of the stem in yellow. The upper right panel shows predicted RNA secondary structures with the left SRP and right arm SRP binding folds highlighted in red. The right lower panels show hypergeometric analyses (Hypergeometric test one-sided) with the enrichment of variants of dominant jumping effect in SRP binding regions (red bar) versus the enrichment of variants of negative jumping effect in SRP binding regions (blue bar). Next to these, sequence logos show DNA nucleotide composition observed in high jumper (top) and low jumper (bottom) haplotypes at select positions.

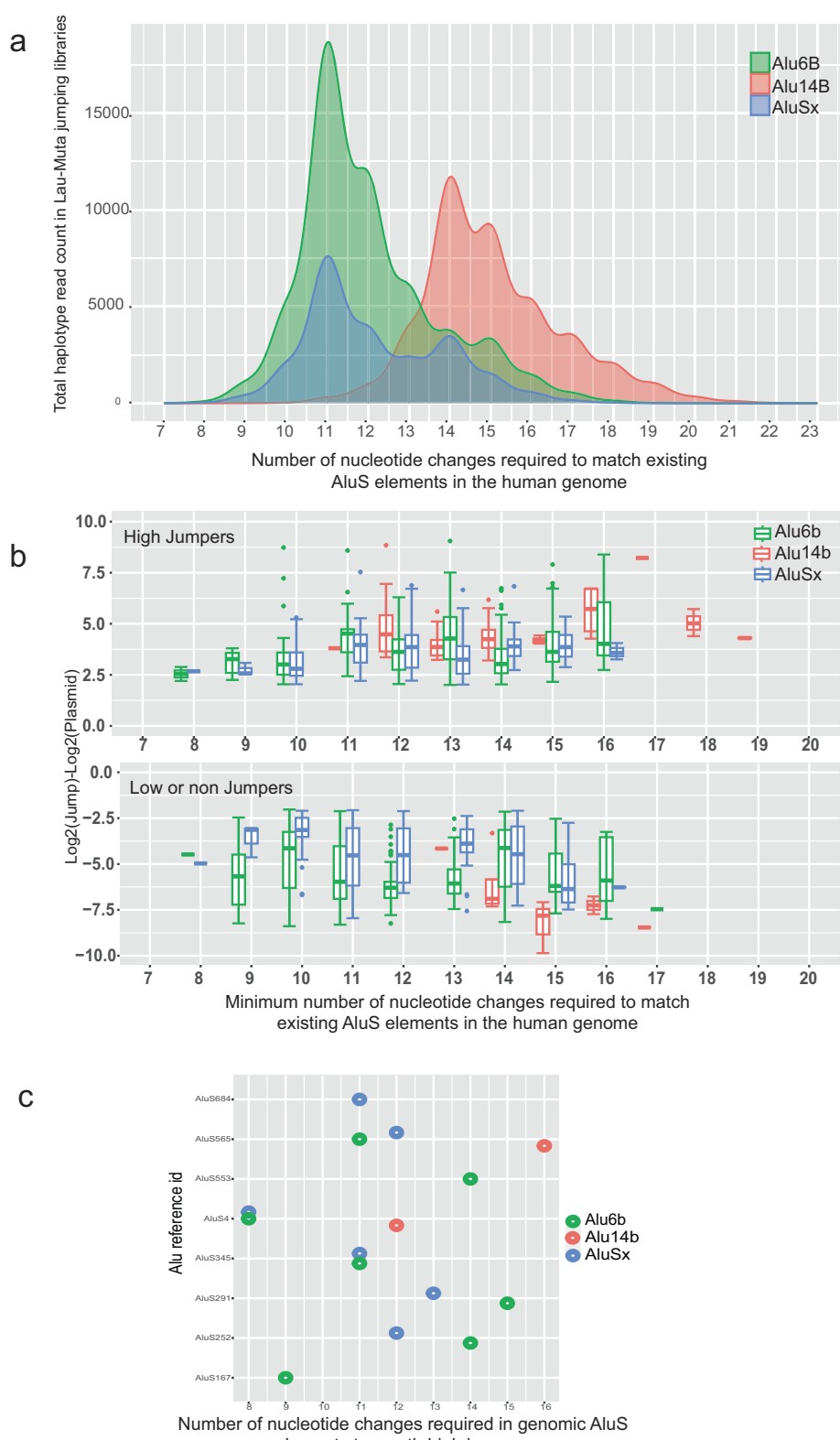

**Fig. 5 | Comparison of endogenous human genome *Alu*S sequences and *Alu*-MPJA haplotypes. a** 852 genomic *Alu*S from the human genome was compared to the haplotypes found in the mutagenized libraries of each element. We detected haplotypes that are at least seven mismatches away from the genomic *Alu*S in *Alu*Sx and *Alu*6B mutagenized libraries and at least eleven mismatches away for *Alu*14B haplotypes. **b** Box and whisker chart showing the mismatch distance of high (top panel) or low (bottom panel) jumpers in the MPJA from genomic *Alu*S sequences. Box and whisker plots include the median line, the box denotes the interquartile range (IQR), whiskers denote the rest of the data distribution, and outliers are denoted by points greater than ±1.5 × IQR. **c** Dot plot showing fifteen identified *Alu*S elements from the human genome (Supplementary Data 8) that require minimum number of nucleotide changes to match with the high jumper haplotypes.

for example, the *Alu*Y family that is still active in the human genome, as well as other types of retrotransposons.

We found that it takes a minimum of one for *Alu*Sx, two for *Alu*6B, and five for *Alu*14B nucleotide changes to increase the jumping potential, most of them residing in SRP binding domains. Here, we limited our analysis to ten mismatches, thus limiting our ability to observe effects for more sequence alterations. To overcome limitations of the error-prone PCR step, a synthetic design of the *Alu* library (e.g., by oligosynthesis) could incorporate widespread changes, including larger deletions or insertions. This could also address the issue of founder effect mutations that we have observed in our mutagenized libraries. In addition, designed *Alu* sequences could specifically test for variant combination effects, i.e., analyzing numerous variant combinations to identify how they interact and affect each other.

There are several critical steps that can affect *Alu* retrotransposition. We used a HeLa-HA cell line that is permissive for Alu activity upon ORF2 supplementation[13,14] and used L1.3 ORF2p for our assays. It will be interesting to score *Alu* retrotransposition activity with other functional ORFs. The *Alu*-RNA with 3′polyA tail folds and forms the secondary structure crucial for binding of SRP9/14 complexes[7]. There is one SRP9/14 binding site in each arm of the *Alu* RNA. This ribonucleoprotein complex recruits L1-ORF2 machinery with reverse-transcriptase and transposon complex and randomly integrates into the genome. Any disruption during this process can lead to abrogation of the jumping potential[12]. We found that mutations in these SRP9/14 domains could affect *Alu*-jumping activity in addition to other mutations in different positions/regions of the *Alu* element. The mutagenesis process we applied was unbiased towards a specific position or domain, as evident in the 5bp-window analysis, thus we could find combinatorial mutations along the full length of the *Alu* element. We validated these results in Supplementary Fig. 12. We found that high-jumping haplotypes have mutations in the SRP9-14 domains, and low-jumping haplotypes have mutations in the right arm structured region. We found that the mutational effect in the right arm supersedes the high-jumping effect of two mutations in the SRP9-14 domains.

In our dataset, non-jumpers could have mutations that disrupt any of the following processes: (1) Transcription of the *Alu* element via RNA Polymerase III; (2) polyA tailing; (3) misfolding of *Alu*-RNA; (4) poor binding of the SRP9/14 complex; (5) inability to recruit the L1-ORF2 machinery. As this assay is primarily designed to identify jumping activity, it is limited in identifying variants that inhibit jumping. Despite that, using stringent conditions, we did detect a small number of variants that abolished jumping, although with lower confidence. Further studies will be required to parse out these other retrotransposition-associated factors in a systematic manner.

We observed a spike in fragments that have >20–25 mismatches only in jumping libraries and not in plasmid libraries. We think this could be due to the inherent nature of our assay system/cells. Since we do not know how *Alu* RNA is processed in these cells, we filtered these fragments from our analyses. However, alternative cells or assay systems may not show this high mismatch number and/or provide an understanding of its cause. Surprisingly, we also observed a small proportion of fragments to be smaller than 200 nucleotides in length in both our jumping libraries and *Alu*-mut plasmid libraries, that could not be explained based on how we processed these sequencing libraries. Of note, it has been reported earlier that the minimal *Alu*-RNA length required for jumping could be less than 200 bp and devoid of its right arm monomer[20], fitting with our finding. Here, we removed these sequences from our analysis, but future MPJA could be used to test what are the critical *Alu* sequences in this left arm monomer that allow it to jump.

MPJA could also be applied to investigate whether *Alu* activity could potentially be involved in evolutionary adaptations or human disease by testing specific somatic mutation haplotypes detected in patients. The haplotype data that we generated could be compared with patient data, such as whole genome sequences from cancer patients, to elucidate if any observed mutations in *Alu* sequences could lead to their reactivation or retrotransposition and consequently increase genomic mutation frequencies. It is also important to consider that *Alu* elements need an active L1 for the retrotransposition machinery, as they are non-autonomous jumpers. L1 is active during early embryonic and brain development, which could affect *Alu* retrotransposition if inherently active *Alu* elements are present in the genome[29,30]. A previous study that individually examined the retrotransposition ability of several *Alu*S, estimated that at least 28 nucleotide changes (10% variation) from the consensus active sequence *Alu*Sx are needed to obtain jumping activity[5,7]. Here, we found that this number can be much lower, with up to seven mismatches for some of the *Alu*S subclasses. We focused on the *Alu*S family for our study, but future studies could be extended to other classes of *Alu* elements in the human genome, for example, the youngest *Alu*Y family elements that are currently active in the genome and have even higher potential to be involved in human disease.

In summary, we provide a high-throughput approach to test the jumping activity of thousands of *Alu* retrotransposons in parallel that could be adopted for other retrotransposons, including DNA transposons. This could facilitate the studies of jumping elements *en masse* in different conditions and be used to address various biological questions related to transposons.

## Methods

### Retrotransposition assay

The *Alu* retrotransposition assay was performed as previously described[5,8,12]. Briefly, HeLa-HA cells (a kind gift from Dr. Astrid Engel, Tulane University)[14] were transfected with lipofectamine LTX reagent using pYa5-neo and pL1-ORF2 or L1 helper (kind gift from Dr. Astrid Engel, Tulane University) at a 1:1 concentration ratio. These HeLa-HA cells need external supplementation of L1.3 ORF2p for retrotransposition. Plasmid pTMO2F3 described earlier was used (Full plasmid sequence in Supplementary Data 1). After 72 h, DMEM with G-418 (500ug/ml) and pen-strep (100 µg/ml) was replaced, and the neomycin selection medium was refreshed every 72 hrs for 2–3 weeks. Colonies were washed with PBS and stained with 0.5% Crystal Violet staining solution containing 10% ethanol and 50% methanol.

### Error-prone PCR and *Alu* library cloning

*Alu*Sx, *Alu*14B, *Alu*6B, and *Alu*h1.1 were synthesized as gene blocks (IDT) and cloned into *Bam*HI-*Aat*II sites in the pYa5-neo vector using the Infusion-cloning kit (Takara) following the manufacturer's protocol. To generate mutagenized libraries with error-prone PCR, we used the GeneMorphII random mutagenesis kit (Agilent) following the manufacturer's protocol. Two independent reactions with 10 ng of plasmid subjected to 25 PCR cycles were pooled and cleaned using the PCR cleanup kit (Qiagen; 28104). The mutagenesis primer (Supplementary Data 1) towards the 3′ of the *Alu* matches the last base in the vector, and therefore, the 3′ of *Alu* was subjected to random mutagenesis. To facilitate subsequent resurrection and retrieval of the jumping *Alu* haplotypes from the genome, mutagenesis was initiated only after the first 27 bases into the 5′ end of the *Alu*. This allowed us to specifically resurrect the retrotransposed and integrated *Alu*-Neo cassette from the genome via a nested PCR amplification strategy (Fig. 1c). Mutagenesis primers had 15 bp overhangs at either side to aid Infusion cloning into *Pst*I-*Aat*II sites of the pYa5-neo vector. Four independent Infusion-cloning reactions were set up and transformed into four different vials of 150 µl each of Stellar competent bacterial cells. Each recovered competent cell vial was used as inoculum for 75 ml of LB broth, which was cultured overnight. Four different

plasmid midi preps (Midi prep kit; Qiagen) were pooled to obtain the *Alu*-Mut library. This procedure was performed separately for each *Alu* element.

## Jumping *Alu*-Mut library retrieval

MPJA was performed on 15 cm cell culture dishes (Corning) in four replicates in two batches and pooled two replicates for genomic DNA isolation together for each batch. After stable selection for 3–4 weeks on G-418, colonies were scrapped from the plate for genomic DNA isolation using a genomic DNA isolation kit (Promega). The spliced and retrotransposed *Alu*-Neo cassette (1 kb) was isolated from genomic DNA using *Alu* recovery and resurrection primers described in Supplementary Data 3. This primer set (PCR-a) allowed to differentiate between the randomly integrated unspliced plasmid DNA product, which is 1.5 kb in length. The 1 kb PCR fragment was then used for nested PCR-b using primers described in Supplementary Data 3. Plasmid mutagenesis libraries were prepared by using PCR-b on the mutagenized plasmid library as the template directly. The amplicon library was then prepped as described above using the Ovation low complexity kit. Libraries were sequenced on Illumina MiSeq with Paired End (PE) 250 bp reads and multiple runs.

## Library sequencing and analysis

*Alu*-Mut plasmid libraries from each batch (*n* = 2) were generated using PCR-based amplification with primer sets described in Supplementary Data 3 using the Ovation low complexity library prep kit (Ovation® Library system for Low complexity samples Part no: 9092-16). Briefly, the PCR amplicon was end-repaired, and forward and reverse diversity adapters were ligated, filled-in, and dual index barcode i5 and i7 primers were used to perform PCR for 10 cycles to generate the *Alu* library fragments (Supplementary Data 10). Libraries were sequenced on Illumina MiSeq with multiple PE 250 bp runs (4 runs for each *Alu* and extra 4 runs for Aluh1.1). PE reads were merged from all the runs and were processed and stitched together to full-length fragments with PEAR[31] and aligned with BWA[32] to *Alu* reference sequences. Custom scripts were used in the analysis for variant calling and filtering available at https://github.com/Ahituv-lab/MPJA. We identified unique haplotypes based on the alignment information stored in BAM format, specifically, we used position and CIGAR fields as well as MD (differences to the aligned reference sequence) and NM (alignment edit distance) tags to identify identical haplotypes and to count their abundance. The edit distance in the NM field was used to filter and to report the minimum number of required edits to the reference sequence. We performed differential activity analysis using DESeq2[17] on unique haplotypes.

## High, low, non-jumper definitions

For all haplotype variants, we compared the fragment counts from two jumping replicates and two plasmid replicates. We applied a stringent cutoff of −log10p > 5 to classify these between high and low jumpers to clean out potential false positives. We use DESeq2[17] to define high jumpers (log2 fold-change > 2, *p* value < 10^−5), low jumpers (log2 fold-change < −2, *p* value < 10^−5, jumping counts > 0) and non-jumpers (log2 Fold-change < −2, *p* value < 10^−5, jumping counts = 0, plasmid counts > 50).

## Inferring variant-level activity effects

To infer the activity effects of individual variants along the different *Alu* elements, we considered every variant (including deletion and insertion) along all haplotypes of a specific experiment. These were included in a matrix with the haplotypes as rows, including their plasmid count, jumping count, and *N* binary columns indicating whether specific variants were associated with that haplotype. We then fit a combined multiple linear regression model of both replicates[15], as

shown in the following equation:

$$\log_2\left(\text{Jumping}_{i,j}\right) \sim \left\{ \begin{matrix} \log_2(\text{Plasmid}_{rep1,j})|i=1 \\ 0|else \end{matrix} \right\} + \left\{ \begin{matrix} \log_2(\text{Plasmid}_{rep2,j})|i=2 \\ 0|else \end{matrix} \right\} + N + \text{offset}$$

## Inferring sliding-window activity effects

To infer the regional activity effects along the different *Alu* elements, we applied a 5 bp sliding-window approach where we use 5 bp windows with an offset of 1 bp along the length of the *Alu* elements. We considered every variant (including deletion and insertion) in each window and considered the combined plasmid count and jumping counts for the center of the window. We then fit a combined multiple linear regression model of both replicates[15].

## Sequence analysis and logos

We analyzed the frequency of nucleotides in short sequence regions, i.e., motifs, by contrasting haplotypes of high and low-jumping activity. For both SRP binding regions (left arm and right arm in each element), we generated logo plots (ggseqlogo R package[33]) with input of SRP left and right binding regions, respectively.

## Hypergeometric test for SRP binding enrichment

To determine whether variants in SRP binding regions are overrepresented or underrepresented in high, low, or non-jumpers, we took the total variants from all *Alu* elements (including all haplotypes) as background. Similar to the gene set enrichment analysis, a significant *p* value obtained from the hypergeometric test suggested that the variants in SRP binding regions are enriched in high jumpers, whereas a non-significant *p* value suggested that variants in SRP binding regions are not enriched in low or non-jumpers. The *p* value was adjusted for multiple testing to control the false discovery rate (Benjamini-Hochberg correction).

## RNA structure prediction

We used the RNA fold web server[25] to predict the secondary structures of single-stranded RNA sequences, for which we first converted *Alu* DNA sequences to RNA. We manually set folding constraints to obtain an *Alu*-like structure, i.e., we are not just reporting the minimum free energy structures, but an *Alu*-RNA-like structure that requires secondary structure folding of the left arm monomer and the right arm monomer separately, with an unstructured middle A-stretch.

## Statistics and reproducibility

No statistical method was used to predetermine sample size since the design of the experiment was based on a synthetic library generated through random mutagenesis in the lab. Sample size and replicates for PCR mutagenesis, jumping assay, library preparation, and sequencing are mentioned in the "Methods" section. Since each *Alu* element library was analyzed separately, we did not exclude any dataset for individual *Alu* elements. The *Alu*h1.1 element library in the scatter plots did not show good correlations between the replicates. Multiple rounds of MiSeqPE250 (total of 8 rounds) were run to increase the sequencing depth. Increasing the sequencing depth did not increase the number of unique haplotypes in any *Alu* library, including for *Alu*h1.1, suggesting that the issue was not the replicates or the sequencing depth (*Alu* haplotype summary tables). We thus removed the *Alu*h1.1 from further analysis. The experiments and sample selection were not randomized. The Investigators were not blinded to allocation during experiments and outcome assessment.

## Reporting summary

Further information on research design is available in the Nature Portfolio Reporting Summary linked to this article.

## Data availability

All sequencing data for this study have been deposited in the GEO database under accession code PRJNA1098913.

## Code availability

The code is available on URL: https://github.com/Ahituv-lab/MPJA.

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

## Acknowledgements

This work was supported in part by the National Human Genome Research Institute (NHGRI) grant numbers UM1HG009408 (N.A.) and UM1HG011966 (M.K., N.A.) and National Institute of General Medical Sciences R01GM142112 (N.A.), Innovative Genomics Institute, IGI-RIDER, and IGI-WIES funding (N.M.). We thank Astrid Engel for providing valuable guidance for retrotransposition assays and sharing vectors and the HeLa cell line.

## Author contributions

N.M. and N.A. conceptualized the study and wrote the manuscript. M.K., J.Z., and N.M. conceived the computational analysis. J.Z. and M.K. performed the computational analysis and wrote scripts. J.Z., M.K., and N.M. performed analyses and created figures. T.J.M. and L.C. provided guidance for the selection of Alu candidates. N.M., A.S., L.D., Y.H., Z.L., K.A., J.S., and S.R. performed the experiments.

## Competing interests

N.M. is a Cofounder and the Chief Scientific Officer for Regel Therapeutics Inc. N.A. is a Cofounder and on the scientific advisory board of Regel Therapeutics Inc. N.A. received funding from BioMarin Pharmaceutical Inc. The remaining authors declare no competing interests.
