## [Peer Review file · Nature Communications]

Massively parallel jumping assay decodes Alu retrotransposition activity

Corresponding Author: Professor Nadav Ahituv

Version 0:

Reviewer comments:

Reviewer #2

(Remarks to the Author)

Matharu et al generate a Alu massively parallel jumping assay (MPJA) based on the introduction by error-prone PCR of mutations into 4 AluS sequences, which are then coupled to an established Alu-G418 retrotransposition reporter. Illumina sequencing is used to calculate the relative incidence of mutations in the AluS mutation library versus sequences recovered after retrotransposition of the G418 reporter. This reveals the relative importance of each position in the AluS sequence in terms of reducing or increasing jumping, as well as how many mutations are required to make a poor AluS jumper into a good AluS jumper (and in what positions). The authors finally consider the AluS haplotypes present in the human genome versus the haplotypes present in the MPJA.

The science looks well done here to me. I liked that the authors explained the key details, and were candid about potential shortcomings in their design. The saturation mutagenesis results were interpreted reasonably. The AluYa5 reporter was a logical starting point. The statistics look robust, using DESeq2 for example to find significant mutations was a good choice it seems given the error-prone PCR can mutate some positions much more than others. Overall, the work is interesting and would make a useful contribution to the field. It could stand to be published without experimental additions. I do however have the following suggestions/questions for clarification (in no particular order):

- 1) line 30: The first sentence of the abstract is a bit chaotic - the meaning is unclear to me. There are millions of retrotransposon copies in the human genome, sure, but not millions of families (and that's what the several refers to). Please clarify.
- 2) line 47: The first sentence of the Introduction is a bit too loose with the statistics. 42% in mammalian genomes ... I'm not sure which genomes those are but it seems high for mammals in general. Perhaps just give the statistic for human (being studied here) or define the species being referred to.
- 3) line 53: Alu can be trans mobilised by an ERV? I do not recall seeing that in the literature but I could simply be ignorant of this finding being made previously. Please double check this.
- 4) The section starting with line 412 should cite PMID:26585388 from Moran.
- 5) line 62: PMID:31575651 provides a more up to date estimate of Alu mobilisation frequency in the human germline (1/40).
- 6) One potentially important point for interpretation is the HeLa sub line being used. PMID:38746229 (currently a preprint) shows that some HeLa lines permit Alu mobilisation readily, whilst others do not. Which is the HeLa line from the Engel lab? This consideration could be mentioned in the Discussion.
- 7) I was confused where the authors say not co-transfecting the Alu-G418 reporter with L1 ORF2p was the negative control. Presumably there is endogenous ORF2p in the HeLa cells being used, and therefore that endogenous ORF2p is present whether additional ORF2p is co-transfected or not?
- 8) Could the authors state in the text which human L1 the ORF2p is generated from? And, in their view, is there evidence to suggest variation in ORF2p could impact their results? And perhaps their results may or may not translate as well to an

experiment conducted with only the native ORF2p available.

9) line 219: what happened with AluH1.1? Removing it due to low reproducibility seems reasonable enough (and I applaud the authors for checking reproducibility in this way and being transparent about decisions) but I would like to know why the reproducibility was low because that reason could affect the other AluS libraries should a third replicate have been performed.

10) For some figures in the review PDF, e.g. Fig. 3a, the labels were too small to read. The presentation could be improved.

11) line 176-178 how many SNVs per base? Extended Data Fig. 5 suggests the mutation pattern was very non-random, as perhaps would be expected. Were there some bases (or mutations of those bases) that were never observed in any of the mutant libraries? Apologies if I missed this information, I could not see it in the text.

12) line 450: some of citations 24-31 are inconclusive or very preliminary and actually there's no citations for papers saying L1 is available in the embryo, brain etc. Perhaps remove some of the less relevant ones amongst citations 24-31 and add one or two for embryo (e.g. PMID:17483097) and brain (e.g. PMID: 31230816)

13) line 351: I found some of this section hard to follow. A simple question ... how many of the AluS copies on the genome would they predict are likely to jump well based on the result from the MPJA?

Geoff Faulkner (University of Queensland)

Reviewer #4

(Remarks to the Author)

Matharu et al. developed a massively parallel jumping assay (MPJA) to probe Alu retrotransposition efficiency of different families in HeLa cells. They first established the retrotransposition assay using AluYa5 following Dewannieux et al. 2003 and constructed mutant library for AluSx, Alu13b, Alu6b, and AluH1.1. They measured the retrotransposition efficiency of different mutants and identified regions within Alu RNA structure that is sensitive to mutations. The approach is innovative and results are convincing. I think this paper is a strong candidate for Nature Communications, and would like to make a few suggestions and comments to help improve the manuscript.

1. The authors claimed to have identified 66,821 jumping haplotypes out of 165,087 haplotypes tested. A major part of the jumping haplotypes was based on Alu14b, which itself is inactive. It wasn't clear to me how the authors assess potential false positive rate. The "jumping library" was normalized to "WT reference". This might be problematic because WT Alu6b and Alu14b are non-jumpers themselves, whereas the WT AluSx is a high jumper. Therefore, the high jumpers defined in the Alu6b and Alu14b libraries might be jumping less than the low-jumpers defined in the AluSx library. It is not clear to me if there is any "spike in" or normalization across different Alu libraries. The bottom line is it remain unclear the sensitivity and specificity of the assay.

2. I would strongly encourage the authors consider performing the same assays in additional cell lines to improve rigor. It is unclear how much of the TE retrotransposition is specific to HeLa cells.

3. I would also encourage the authors to validate some of the key results from the high throughput assay by using classic jumping assays.

4. A key finding that Alu stem-loop RNA structures mediate association with SRP and retro-transposition is interesting. They imply that mutations in the stem-associated regions likely act by modifying this mechanism, however, they do not provide any follow-up or experimental support for this claim. They should demonstrate that their mutations act via this mechanism.

5. Mutations from the mutagenesis PCR process are not random, as the authors admitted. For example, it looks like every AluSX-Mut sequence has the same mutations in the SRP9-14 binding region, which might bias the results from Figure 4, which shows the importance of this domain for jumping ability, as we don't know how the other variants would interact with a wild-type SRP9-14 binding region.

6. The section of Figure 4 that highlights differences in SRP9/14 motifs between high and low jumpers is unclear to me. For instance, it appears that there are no differences between high and low jumpers at bases 24 and 34 (which they highlight as their "selected positions" in B (Alu14B), as well as bp 162 in C (AluSx). This is also true of several parts of extended data fig. 9.

7. The correlation values for Extended Data Figure 4b are presented in the main text as being similarly strong between the jumping haplotypes for Alu6B and AluSX, however the plot for AluSX shows some clear sub-populations that deviate from the center line. Do the authors have any thoughts on these subpopulations?

8. The authors say "as they need to hijack transposition machinery from other retrotransposons (i.e., LINES/LTRs) to jump" – please provide reference since I don't know Alu can use LTR mechanism to jump.

9. The authors say "AluY, the youngest lineage (~10 myo) with almost all AluY elements thought to be active.". AluY is present in old world monkey, thus the oldest elements can be at least 25myo. Only the most recent ALuY subfamilies are active in the human genome.

Version 1:

Reviewer comments:

Reviewer #2

(Remarks to the Author)

The authors have addressed all of my comments, thank you.

Geoff Faulkner (University of Queensland)

Reviewer #4

(Remarks to the Author)

I thank the authors for addressing my comments. I do not have further questions.

We want to thank the reviewers for their comments that have significantly improved our revised manuscript. Below we provide a point-by-point response for these comments and have also used blue font in the revised manuscript to highlight changes we made to address reviewers comments.

Reviewer #2

Matharu et al generate a Alu massively parallel jumping assay (MPJA) based on the introduction by error-prone PCR of mutations into 4 AluS sequences, which are then coupled to an established Alu-G418 retrotransposition reporter. Illumina sequencing is used to calculate the relative incidence of mutations in the AluS mutation library versus sequences recovered after retrotransposition of the G418 reporter. This reveals the relative importance of each position in the AluS sequence in terms of reducing or increasing jumping, as well as how many mutations are required to make a poor AluS jumper into a good AluS jumper (and in what positions). The authors finally consider the AluS haplotypes present in the human genome versus the haplotypes present in the MPJA.

The science looks well done here to me. I liked that the authors explained the key details, and were candid about potential shortcomings in their design. The saturation mutagenesis results were interpreted reasonably. The AluYa5 reporter was a logical starting point. The statistics look robust, using DESeq2 for example to find significant mutations was a good choice it seems given the error-prone PCR can mutate some positions much more than others. Overall, the work is interesting and would make a useful contribution to the field. It could stand to be published without experimental additions. I do however have the following suggestions/questions for clarification (in no particular order):

We thank the reviewer for being supportive of this work and for their comments which have greatly improved the revised version of our manuscript.

1) line 30: The first sentence of the abstract is a bit chaotic - the meaning is unclear to me. There are millions of retrotransposon copies in the human genome, sure, but not millions of families (and that's what the several refers to). Please clarify.

We thank the reviewer for pointing this out. To clarify, we have now reframed the sentence according to the suggestion:

Line 30: 'The human genome contains millions of copies of retrotransposons that are silenced but many of these copies have potential to become active if mutated, having phenotypic consequences, including disease.'

2) line 47: The first sentence of the Introduction is a bit too loose with the statistics. 42% in mammalian genomes ... I'm not sure which genomes those are but it seems high for mammals in general. Perhaps just give the statistic for human (being studied here) or define the species being referred to.

We have now reframed the first sentence of the Introduction according to the suggestion:

Line 44: *'An estimated 42% of the human genome consists of retrotransposable elements that 'copy and paste' in genomes via RNA mediated transposition'* and cited the following article for this: International Human Genome Sequencing Consortium. Initial sequencing and analysis of the human genome. *Nature* **409**, 860–921 (2001). <https://doi.org/10.1038/35057062>

3) line 53: Alu can be trans mobilised by an ERV? I do not recall seeing that in the literature but I could simply be ignorant of this finding being made previously. Please double check this.

We thank the reviewer for pointing this out we have now reframed the sentence with a citation: *Line 49-50*:

'Alu elements are non-autonomous retrotransposons, as they borrow transposition machinery from other retrotransposons (i.e., L1-ORF2p) to jump³'.

Reference 3: Deininger, P. Alu elements: Know the SINEs. *Genome Biology* vol. 12 <https://doi.org/10.1186/gb-2011-12-12-236> (2011).

4) The section starting with line 412 should cite PMID:26585388 from Moran.

We thank the reviewer for suggesting the appropriate reference. We have now added this citation by A.J. Doucet et al. (DOI: 10.1016/j.molcel.2015.10.012) in the revised version at Line 490 (reference 7):

'A previous study that individually examined the retrotransposition ability of several AluS, estimated that at least 28 nucleotide changes (10% variation) from the consensus active sequence AluSx are needed to obtain jumping activity^{5,7}'.

5) line 62: PMID:31575651 provides a more up to date estimate of Alu mobilisation frequency in the human germline (1/40).

We thank the reviewer for suggesting the updated reference. We have now cited Feusier J. et al. (DOI: 10.1101/gr.247965.118) in the revised version at line 59-60 (Reference 6):

'The human genome is predicted to have 852 intact functional Alu elements with thousands of copies that could be competent to jump⁵ and it is estimated that a new Alu insertion happens in the human genome every 40 live births⁶'.

6) One potentially important point for interpretation is the HeLa sub line being used. PMID:38746229 (currently a preprint) shows that some HeLa lines permit Alu mobilisation readily, whilst others do not. Which is the HeLa line from the Engel lab? This consideration could be mentioned in the Discussion.

We thank the reviewer for suggesting the updated reference. We have now cited the manuscript by J.B. Moldovan et al., that was published by NAR (<https://doi.org/10.1093/nar/gkae448>) in the meantime, in the revised version along with the discussion on HeLa sub-lines used for jumping assays. We have used HeLa-HA cells (kind gift from Astrid Engel's lab). We have added this information at appropriate places in the manuscript: under the Results section "*Retrotransposition*

assay optimization". Line 115. Also in Discussion Line 443 and Methods section Line 510.

7) I was confused where the authors say not co-transfecting the Alu-G418 reporter with L1 ORF2p was the negative control. Presumably there is endogenous ORF2p in the HeLa cells being used, and therefore that endogenous ORF2p is present whether additional ORF2p is co-transfected or not?

We have performed these Alu mobilization assays in HeLa-HA cells as described by J.B. Moldovan et al. (PMID:38746229), which needs ORF2 co-transfection to score for retrotransposition activity. We have now added this in methods section line 510:

"These HeLa-Ha cells need external supplementation of L1.3 ORF2p for retrotransposition of Alu elements. Plasmid pTMO2F3 described earlier was used here".

8) Could the authors state in the text which human L1 the ORF2p is generated from? And, in their view, is there evidence to suggest variation in ORF2p could impact their results? And perhaps their results may or may not translate as well to an experiment conducted with only the native ORF2p available.

We thank reviewer for asking this question. We have used L1.3 ORF2p in the expression plasmid named: pORF2 derived from pTMO2F3 backbone, but without a FLAG tag, as described by J.B. Moldovan et al. (PMID:38746229) and J.V. Moran et al. (PMID: 8945518). We have now added this with a full sequence of this plasmid containing ORF2 in Supplementary Table 1 under the plasmid sequence section.

It has been shown that a mutation in the reverse transcriptase domain in this wild type L1-ORF2p abolishes the retrotransposition activity in HeLa-HA cells (PMID:38746229). We agree that it will be very interesting to test our assay with other ORFs, but feel it is beyond the scope of this article since we needed the standard functional transposase for this assay. We have added the following text to the discussion to mention this option.

Line 444: 'We used a HeLa-HA cell line that is permissive for Alu activity upon ORF2 supplementation^{13,14} and used L1.3 ORF2p for our assays. It will be interesting to score Alu retrotransposition activity with other functional ORFs.'

9) line 219: what happened with Aluh1.1? Removing it due to low reproducibility seems reasonable enough (and I applaud the authors for checking reproducibility in this way and being transparent about decisions) but I would like to know why the reproducibility was low because that reason could affect the other AluS libraries should a third replicate have been performed.

This is a great comment and one we wish we could have an easy answer for. In the scatter plots we did not get good correlations between the replicates for Aluh1.1 despite having almost similar unique haplotypes for other Alu elements tested here. First, we thought it could be due to the sequencing depth and performed a couple of extra rounds of sequencing for all Alu libraries including the Aluh1.1. Increasing

sequencing depth did not increase the number of unique haplotypes in any Alu library including for Aluh1.1 suggesting that the issue was not the replicates or the sequencing depth (Alu haplotype summary table). We thus think it could have something to do with the Aluh1.1 biology because somehow we got way higher mutation rates in Aluh1.1. We do think that forward designing of the Aluh1.1 mutagenization with an oligo synthesis library could address this issue in the future as mentioned in the discussion section.

10) For some figures in the review PDF, e.g. Fig. 3a, the labels were too small to read. The presentation could be improved.

We thank the reviewer for this suggestion. We checked figures and increased the font size of the labels (e.g. Fig3a).

11) line 176-178 how many SNVs per base? Extended Data Fig. 5 suggests the mutation pattern was very non-random, as perhaps would be expected. Were there some bases (or mutations of those bases) that were never observed in any of the mutant libraries? Apologies if I missed this information, I could not see it in the text.

We thank the reviewer for this comment. We asked if there were certain positions where mutations weren't observed in Alu plasmid libraries?

We found non-mutated bases/positions: in AluSx: position:1,2,3,4 and 5, in Alu6b: positions 1,3,4 and 5 in Alu 14b: 1,2,3,4 and 5 in Alu h11: 1, 2, 3, 4, 5, 9, 10, 283 and 284. We also checked, there's no position that is 100% mutated either, basically where we found all SNV and INDELS. This is expected due to random mutagenesis.

We have now added this info in the text line 179 under section "Alu-MPJA saturation mutagenesis:

Line 179: 'We also observed that nucleotide positions 1-5 in AluSx, Alu6b and Alu14b libraries were never mutated. This could be attributed to our random mutagenesis strategy that includes a 5' primer overlapping up to 20 terminal nucleotides (**Extended data Fig 10**). For Aluh1.1, base positions 1-5, 9, 10, 283 and 284 were never mutated. We also observed that there was no nucleotide position that was 100% mutated, i.e. created all possible mutations for that position including SNV and INDELS.'

12) line 450: some of citations 24-31 are inconclusive or very preliminary and actually there's no citations for papers saying L1 is available in the embryo, brain etc. Perhaps remove some of the less relevant ones amongst citations 24-31 and add one or two for embryo (e.g. PMID:17483097) and brain (e.g. PMID: 31230816)

We thank the reviewer and addressed these comments in the revised version removing some citations and replacing them with others. This is now in line 485 in the revised version with only two relevant citations as suggested by the reviewer:

Line 487: "It is also important to consider that Alu elements need an active L1 for the retrotransposition machinery, as they are non-autonomous jumpers. L1 is active during early embryonic and brain development, which could affect Alu retrotransposition if inherently active Alu elements are present in the genome."

13) line 351: I found some of this section hard to follow. A simple question ... how

many of the AluS copies on the genome would they predict are likely to jump well based on the result from the MPJA?

We thank the reviewer for giving us this opportunity to improve this results section. In the current genome we do not expect any Alu copies to jump unless 8-16 mutations are incorporated that could render them active. In our assay, we found that some of the currently present Alu elements in the human genome are 8-16 mutations away from becoming active Alu elements. We have clarified this in the text and revised this section at several places.

Line 376: AluS family of elements in the human genome are inactive due to sequence changes. We next set out to quantify how many mutations could lead to the activation of current AluS elements in the human reference genome if mutated.

Line 381: We first aligned our AluS jumping haplotypes to the subset of AluS sequences in the human genome that have an intact 280bp core region, finding that none of these mutagenized haplotypes have an exact match with these elements. This was expected as previously reported by testing 27 AluS elements individually, that no full length AluS is active in a retrotransposition assay when it is 10% divergent (28 nucleotide changes) from the consensus active AluSx sequence

Line 390: From our MPJA study, we found that the closest jumping mutagenized haplotypes for Alu6B or AluSx are seven mismatches away from endogenous AluS and ten mismatches for Alu14B (Fig. 5a-b).

Line 393: We also looked at how many AluS elements in the human genome could potentially become jumpers upon n number of mismatches.

Line 396: We found up to fifteen high jumping haplotypes in our library similar to the eight distinct Alu elements from the genome that are 8-16 mismatches away from being high jumpers (Fig. 5c). In our assay we found that some of the currently present Alu elements in the human genome could become active if they incorporate 8-16 mutations.

Reviewer #4

Matharu et al. developed a massively parallel jumping assay (MPJA) to probe Alu retrotransposition efficiency of different families in HeLa cells. They first established the retrotransposition assay using AluYa5 following Dewannieux et al. 2003 and constructed mutant library for AluSx, Alu13b, Alu6b, and Alu1.1. They measured the retrotransposition efficiency of different mutants and identified regions within Alu RNA structure that is sensitive to mutations. The approach is innovative and results are convincing. I think this paper is a strong candidate for Nature Communications, and would like to make a few suggestions and comments to help improve the manuscript.

We thank the reviewer for being supportive of this work and for their comments that have greatly improved the revised version of this manuscript.

1. The authors claimed to have identified 66,821 jumping haplotypes out of 165,087 haplotypes tested. A major part of the jumping haplotypes was based on Alu14b, which itself is inactive. It wasn't clear to me how the authors assess potential false positive rate. The "jumping library" was normalized to "WT reference". This might be problematic because WT Alu6b and Alu14b are non-jumpers themselves, whereas the WT AluSx is a high jumper. Therefore, the high jumpers defined in the Alu6b and Alu14b libraries might be jumping less than the low-jumpers defined in the AluSx library. It is not clear to me if there is any "spike in" or normalization across different Alu libraries. The bottom line is it remain unclear the sensitivity and specificity of the assay.

We thank this reviewer for giving us the opportunity to clarify this point. There were a total of 66,821 haplotypes detected in the jumping libraries out of 165,087 detected in the plasmid libraries. We analyzed each Alu element library separately. We applied a very stringent p-value cutoff of 10^{-5} to classify these between high and low jumpers, for each Alu element library separately, to limit the number of potential false positives. Consequently, most of the Alu14b haplotypes did not cross that confidence filter and only around 0.1% of them passed this filter. For AluSx, it was 1.3% and 1% of the haplotypes from Alu6B passed that threshold. We have now added this info in **Supplementary Table 12**. Our study provides correlation of jumping activities or nucleotide variants per each individual Alu element, therefore we benchmark each Alu element with its own wild-type version and interpret results within each Alu element. As shown in Fig2c, Alu6B and Alu14B are inherently inactive when compared with AluSx WT reference but still have some residual jumping activity as also evident in the colony assays (**Extended Data Fig. 3**). Due to this reason, we did not carry out any cross Alu comparisons, as this would have led to assess only AluSx haplotypes be the high jumpers due to its inherent retrotransposition activity. To clarify this better, we have now added an **Extended Data Figure 11** where we have added the Volcano plots without normalizing to the wild type fold change. We clarified this at several places in the text and in methods section.

Line 266: *'We next set out to identify Alu haplotypes that lead to a significant increase in retrotransposition compared to their reference wildtype Alu sequence, by measuring differential enrichment in Alu-Mut jumping vs Alu-Mut plasmid libraries.'*

Line 279: *'Due to the difference in their inherent jumping activity, we compared haplotypes within each Alu library and not across them. We also generated volcano plots for the haplotypes with non-normalized results to the Alu reference WT sequence (Extended data Figure 11).'*

Line 427: *'We tested different unique haplotypes for each Alu element and compared them with their reference wild type version to record their jumping potential.'*

2. I would strongly encourage the authors consider performing the same assays in additional cell lines to improve rigor. It is unclear how much of the TE retrotransposition is specific to HeLa cells.

We thank the reviewer for highlighting the importance of cell lines here. This exact comment has been dealt with in a separate article (PMID:38746229, John B Moldovan et al.) that was recently published in *Nucleic Acid Res.* and now cited in our revised manuscript. In this article, they tested 4 different cell lines for Alu retrotransposition activity and found HeLa-HA and CCL2 to be equally permissive for Alu activity while other cell lines were not. We used HeLa-HA in this article which is one of the standard cell lines of choice for all previous literature reporting on Alu-retrotransposition assays. We think testing in other cell lines will most likely come up with similar results as already reported in PMID:38746229. We have added the following text in the revised manuscript in the Discussion section:

Line 444:

'We used a HeLa-HA cell line that is permissive for Alu activity upon ORF2 supplementation as described earlier^{13,14} and used L1.3 ORF2p for our assays. It will be interesting to score Alu retrotransposition activity with other functional ORFs.'

3. I would also encourage the authors to validate some of the key results from the high throughput assay by using classic jumping assays.

We thank the reviewer for encouraging us to perform the validation assay. We have now performed the validation assay for 2 high jumpers and 2 low jumpers using a classic colony forming unit assay (**Extended data figure 12**) and added the following text:

Line 356: *'We validated these results using a standard retrotransposition colony assay by testing two high and two low jumping haplotypes for AluSx, having mutations in their SRP domains, finding similar results to the MPJA (Extended data Figure 12).'*

4. A key finding that Alu stem-loop RNA structures mediate association with SRP and retro-transposition is interesting. They imply that mutations in the stem-associated regions likely act by modifying this mechanism, however, they do not provide any follow-up or experimental support for this claim. They should demonstrate that their mutations act via this mechanism.

We thank the reviewer for raising this comment. Alu elements are poorly annotated with respect to the structural and positional information. The only well described structures in AluRNA are the stem loop SRP9-14 domains. It's known that these

domains are important for forming SRP complexes that support Alu activity. It was an interesting observation that we found mutations in these domains affecting Alu jumping activity along with other mutations in different positions/regions of the Alu element. The mutagenesis process we applied was unbiased towards any specific position or domain, as evident in the 5bp-window analysis. As mentioned in the comment above, we validated these results in **Extended Data Figure 12** where we tested high jumping haplotypes that have mutations in SRP9-14 domains while low jumping haplotypes have mutations in the right arm structured region that seem to supersede the high jumping effect of the mutation in SRP9-14 domains. In addition to adding these results, we have also added the following text in the Discussion:

Line 447: 'The *Alu*-RNA with 3'polyA tail folds and forms the secondary structure crucial for binding of SRP9/14 complexes⁷. There is one SRP9/14 binding site in each arm of the Alu RNA. This ribonucleoprotein complex recruits the L1-ORF2 machinery with reverse-transcriptase and transposon complex and randomly integrates into the genome. Any disruption during this process can lead to abrogation of the jumping potential¹². We found that mutations in these SRP9/14 domains could affect Alu jumping activity in addition to other mutations in different positions/regions of the Alu element. The mutagenesis process we applied was unbiased towards a specific position or domain, as evident in the 5bp-window analysis, thus we could find combinatorial mutations along the full length of the Alu element. We validated these results in **Extended Data Figure 12**. We found that high jumping haplotypes have mutations in SRP9-14 domains and low jumping haplotypes have mutations in the right arm structured region. We found that the mutational effect in the right arm supersedes the high jumping effect of two mutations in the SRP9-14 domains.

5. Mutations from the mutagenesis PCR process are not random, as the authors admitted. For example, it looks like every AluSX-Mut sequence has the same mutations in the SRP9-14 binding region, which might bias the results from Figure 4, which shows the importance of this domain for jumping ability, as we don't know how the other variants would interact with a wild-type SRP9-14 binding region.

We thank the reviewer for pointing this out and giving us an opportunity to address this. We added **Extended Data Figure 10** and **Supplementary Table 11** to clarify this point further which was not evident in the earlier version of the manuscript. We think these changes have greatly improved the interpretations and clarified our result better.

We agree that the PCR mutagenesis strategy is not entirely random and shows an imprint of the specific enzymes and buffer conditions. However, we would like to take this opportunity to clarify that we found many different SNVs in the SRP9-14 binding region of AluSx-Mut sequences in the plasmid library, suggesting that we created different haplotypes here. After jumping, these haplotypes underwent retrotransposition and, depending upon their ability to jump, were enriched or depleted. To further clarify this, we have now added the mutation density Lollipop plots for the jumping Alu-Mut libraries in **Extended Figure 10** along with the following text:

Line 213: 'We observed that highly enriched SNVs in the *Alu*-Mut plasmid library, either due to PCR bias or founder effect, were not necessarily enriched in the *Alu*-Mut jumping library, confirming that our measurements are focused on jumping activity (**Fig. 2c, Extended Data Fig 10**). As mentioned earlier, the PCR mutagenesis strategy is non-random and can lead to founder mutation effects. However, we found many different SNVs in the SRP9-14 binding region of *Alu*Sx-Mut sequences in the plasmid library, suggesting that we created different haplotypes at 'founder positions' although with different frequencies. After jumping, these haplotypes underwent retrotransposition and depending upon their ability to jump, these haplotypes were enriched or depleted in the *Alu*-Mut jumping libraries.'

Line 249: 'We found that the aforementioned highly enriched SNVs in the plasmid libraries, which likely appeared due to PCR bias and founder effects during library generation, were not significantly enriched in *Alu* jumping libraries (with $\pm 2 \log_2 FC$, $p\text{-value} < 10^{-5}$), suggesting that they have a minimal effect on determining jumping activity (**Fig. 2c, Extended Data Fig 10**).'

In addition, we discuss the issue of 'founder effect' mutations as one of the limitations of error-prone PCR strategy leading to the combinatorial effect of other mutations being masked. After subjecting haplotypes to the retrotransposition jumping assay the founder mutation frequency was drastically different than in the original plasmid library. Their frequency clearly depends on the fact that certain haplotypes were capable of retrotransposition or not, as demonstrated in **Extended Data Figure 10** in the revised version of the manuscript. We discuss the resulting limited combinatorial representation in the Discussion section as one of the caveats of random mutagenesis. We propose that designing oligosynthetic libraries with controlled mutation frequencies could address this issue in future studies. We have added the following text to the discussion:

Line 436: 'To overcome limitations of the error-prone PCR step, a synthetic design of the *Alu* library (e.g., by oligosynthesis) could incorporate widespread changes including larger deletions or insertions. This could also address the issue of founder effect mutations that we observed in our mutagenized libraries. In addition, designed *Alu* sequences could specifically test for variant combination effects, i.e. analyzing numerous variant combinations to identify how they interact and affect each other.'

Specifically for **Figure 4**, we compared the nucleotide changes that occur in high versus low jumpers. For example, position 153 shows different mutations in both low and high jumpers that were not the founder SNV mutation. There are many haplotypes observed (with different SNVs) at position 153 suggesting that there was no bias that was created/selected during jumping because of the founder mutation in the plasmid library. We also revised the logo plot in **Figure 4** to highlight this, such that other variants are represented. In addition, to thoroughly address this concern we have now added **Supplementary Table 11** reporting the frequency of each nucleotide in each SRP domain in the left and right arm separately for low and high jumping *Alu* haplotypes. We have added this in the Results text:

Line 349 'We compared the nucleotide changes that occur in high versus low jumpers. For example, position 153 shows different mutations in both low and high jumpers that were not the founder SNV mutation of the plasmid library. The many

haplotypes observed (with different SNVs) at position 153 suggest that there was no bias from the founder mutation in the plasmid library during jumping. We report the frequency of each nucleotide in each SRP domain in the left and right arm separately for low and high jumping Alu haplotypes (**Supplementary Table 11**).'

6. The section of Figure 4 that highlights differences in SRP9/14 motifs between high and low jumpers is unclear to me. For instance, it appears that there are no differences between high and low jumpers at bases 24 and 34 (which they highlight as their "selected positions" in B (Alu14B), as well as bp 162 in C (AluSx). This is also true of several parts of extended data fig. 9.

We thank the reviewer for this comment, and think this has given us opportunity to depict our results with more clarity. To address this comment clearly, we have now added **Supplementary Table 11** which reports the frequency of nucleotides in each SRP domain in the left and right arm separately for low and high jumping Alu haplotypes, clearly depicting the frequency of each nucleotide position.

The highlighted positions are the ones where there is a difference in observed frequency of certain nucleotides between high jumpers and low jumpers haplotypes. For example, high jumpers in Alu 14B have T>C variants at position 24 and A>G variants at position 34. However T at position 24 and A at position 34 are both observed in high and low jumpers where mutations in some other domains could be affecting the jumping activity. To clarify this better we have revised **Figure 4** and added more information in the results section. We have revised the logo plots figure to highlight this, such that other variants are also represented.

7. The correlation values for Extended Data Figure 4b are presented in the main text as being similarly strong between the jumping haplotypes for Alu6B and AluSX, however the plot for AluSX shows some clear sub-populations that deviate from the center line. Do the authors have any thoughts on these subpopulations?

The deviant points from the centre line were not enough to affect the overall correlation of the plot substantially. While we cannot comment on the nature of these dispersed subpopulations, further analysis showed that due to their large variance between replicates, these are filtered out from subsequent analyses. Specifically, the DESeq2 analysis that we performed to calculate the fold changes among the replicates estimates the dispersion for each haplotype which quantifies how much a haplotype's read counts vary among replicates beyond what would be expected from the Null model (i.e., Negative Binomial approximating the over dispersed Poisson sampling process). We checked this dispersion and these haplotypes would not pass our p-value filters and hence were filtered out in further analysis.

8. The authors say "as they need to hijack transposition machinery from other retrotransposons (i.e., LINES/LTRs) to jump" – please provide reference since I don't know Alu can use LTR mechanism to jump.

We thank the reviewer for pointing this out. We have now reframed the sentence to read:

Line 49: 'Alu elements are non-autonomous retrotransposons, as they borrow transposition machinery from other retrotransposons (i.e., L1-ORF2p) to jump.'

9. The authors say "AluY, the youngest lineage (~10 myo) with almost all AluY elements thought to be active." AluY is present in old world monkey, thus the oldest elements can be at least 25myo. Only the most recent ALuY subfamilies are active in the human genome.

We thank the reviewer for this suggestion. We have now corrected this sentence accordingly.

Line 56: "AluY, is present in old world monkeys at least ~25myo ago. Some of the recent (~10 myo) AluY elements are thought to be active in the human genome.